# Cytostatic Activity of Sanguinarine and a Cyanide Derivative in Human Erythroleukemia Cells Is Mediated by Suppression of c-MET/MAPK Signaling

**DOI:** 10.3390/ijms24098113

**Published:** 2023-04-30

**Authors:** Xinglian Xu, Lulu Deng, Yaling Tang, Jiang Li, Ting Zhong, Xiaojiang Hao, Yanhua Fan, Shuzhen Mu

**Affiliations:** 1State Key Laboratory of Functions and Applications of Medicinal Plants, Guizhou Medical University, 3491 Beijin Road, Guiyang 550014, China; 2The Key Laboratory of Chemistry for Natural Products of Guizhou Province and Chinese Academy of Sciences, 3491 Beijin Road, Guiyang 550014, China; 3Kunming Institute of Botany, Chinese Academy of Sciences (CAS), Kunming 650201, China

**Keywords:** leukemia, sanguinarine, c-MET, MAPK, PI3K/mTOR, STAT3

## Abstract

Sanguinarine (**1**) is a natural product with significant pharmacological effects. However, the application of sanguinarine has been limited due to its toxic side effects and a lack of clarity regarding its molecular mechanisms. To reduce the toxic side effects of sanguinarine, its cyanide derivative (**1a**) was first designed and synthesized in our previous research. In this study, we confirmed that **1a** presents lower toxicity than sanguinarine but shows comparable anti-leukemia activity. Further biological studies using RNA-seq, lentiviral transfection, Western blotting, and flow cytometry analysis first revealed that both compounds **1** and **1a** inhibited the proliferation and induced the apoptosis of leukemic cells by regulating the transcription of c-MET and then suppressing downstream pathways, including the MAPK, PI3K/AKT and JAK/STAT pathways. Collectively, the data indicate that **1a**, as a potential anti-leukemia lead compound regulating c-MET transcription, exhibits better safety than **1** while maintaining cytostatic activity through the same mechanism as **1**.

## 1. Introduction

Sanguinarine (**1**) (Figure 1) is an active natural product isolated from many medicinal plants [1]. Its pharmacological effects include antitumor, antibacterial, antiviral, and anti-inflammatory effects [2,3,4,5]. Antitumor studies on sanguinarine have mainly focused on several major human cancers, including lung, breast, and bladder cancer, and other solid tumors [6]. Previous studies indicated that the toxicity of sanguinarine is the main factor limiting its clinical application in cancer therapy [7,8]. Therefore, it is necessary to perform structural optimization to reduce toxicity.

To improve the cytostatic activity of **1** and reduce its toxic effects on normal cells, in our previous work we carried out structural modification and assessed the anti-leukemia activities of its derivatives [9]. We found that the activity of **1** was mainly dependent on the following structural features: the fusion of the B/C ring, hydroxylation patterns, N-methyl substitutions, and planar configuration [10]. Most notably, the cyanide derivative (**1a**) (Figure 1) showed comparable anti-leukemic activity and lower normal hepatic stellate cell cytotoxicity than **1**; however, the molecular mechanisms of the anti-leukemic effect remain to be explored.

According to previous research, **1** exerts its cytostatic effects mainly by inducing the apoptosis of tumor cells through regulatory proteins and signaling pathways, including the mitogen-activated protein kinase (MAPK) [11], PI3K/AKT [12], JAK2/STAT3, and P53 pathways [13,14]. However, it is not clear how these signaling pathways and proteins are regulated by **1**. Therefore, it is crucial to further explore the underlying anticancer mechanism or targets of **1**. Upstream proteins such as c-MET, EGFR, and HER2 are involved in compound-**1**-mediated regulation of the MAPK, PI3K/AKT, and STAT3 signaling pathways [15,16,17]. The c-MET receptor, a receptor tyrosine kinase (RTK), is the cell surface receptor for hepatocyte growth factor (HGF) encoded by the MET proto-oncogene [18]. It plays essential roles in controlling a number of critical cellular processes [19]. Aberrant activation of c-MET can lead to tumor progression, invasive growth, angiogenesis, metastasis, and resistance to therapies, and plays a vital role in the treatment of tumors, especially hepatocellular carcinoma; therefore, c-MET has become a popular target for tumor treatment [20].

To date, there are no reports about the effect of **1** on c-MET. In this work, we investigated the effects of **1** and its derivative **1a** on c-MET at the transcript and protein levels. This study might reveal a new antitumor mechanism and provide a solid theoretical basis for the development and utilization of **1** in tumor therapy.

## 2. Results and Discussion

### 2.1. Compounds **1** and **1a** Inhibited the Proliferation of Human Erythroid Leukemia Cells

Leukemia is a class of clonal malignant diseases involving the abnormal expansion of hepatic stellate cells [21]. At present, the treatment of leukemia is still a major challenge, and most of the drugs used clinically for the treatment of leukemia have various degrees of adverse effects [22]. There is an urgent need for effective lead compounds for the treatment of leukemia. Compound **1** is a natural product isolated from *Zanthoxylum nitidum* (Roxb.) DC with significant inhibitory activity against leukemia cells but obvious cytotoxicity against normal cells. To improve the cytostatic activity of **1** and reduce its toxic effects, a series of derivatives were synthesized (Appendix A), and the cytostatic activity of the derivatives including **1a** was screened (Appendix A). Notably, the cytostatic activity of **1a** was comparable to that of **1** (Figure 2A), but **1a** was less cytotoxic than **1** to normal LX-2 hepatic stellate cells (Figure 2B). To further investigate whether the molecular mechanisms of compound **1a** were the same as those of the lead compound (compound **1)**, 0.75 μM, 1.50 μM, and 3.00 μM, and 1.00 μM, 2.00 μM, and 4.00 μM were separately selected as experimental concentrations for **1** and **1a**.

### 2.2. The Effect of **1** and **1a** on Cell Apoptosis

The induction of apoptosis is considered to be one of the major mechanisms of **1′s** inhibition of tumor cells [2]. To assess the effect of **1** on the apoptosis of HEL cells, cells were stained with Hoechst 33258 [23]. As shown in Figure 3A, the intensity of bright blue fluorescence was enhanced with increasing concentrations after treatment with **1** and **1a**, and the fluorescence spots were evenly distributed. Compared with that of the blank control group, the fluorescence intensity of the treatment group was significantly higher. Meanwhile, TUNEL staining showed that treatment with **1** and **1a** significantly induced cell apoptosis (Figure 3B) [24]. The above results demonstrate that both **1** and **1a** could induce the apoptosis of HEL cells in a dose-dependent manner.

To further explore the effect of **1** and **1a** on the viability of HEL cells, an Annexin V-FITC/propidium iodide (PI) double-staining apoptosis kit was used to measure the apoptosis rate of HEL cells [25]. HEL cells were treated with different concentrations of the compounds, and the staining results showed that the compounds increased the apoptosis rate of HEL cells in a dose-dependent manner (Figure 3C). Then, Western blotting was used to analyze the changes in apoptotic proteins after the treatment of HEL cells with the compounds [26]. The results showed that the compounds increased the levels of cl-PARP protein in a dose-dependent manner (Figure 3D). PARP is a DNA repair enzyme that plays an important role in DNA damage repair and apoptosis, and PARP shearing is considered to be an important indicator of apoptosis [27]. Overall, these results clearly confirmed that **1** and **1a** can induce HEL cell apoptosis.

### 2.3. Compounds **1** and **1a** Exerted Pharmacological Antitumor Effects by Blocking MAPK Signaling

To elucidate the molecular mechanisms of **1** and **1a** in leukemia, RNA-seq analysis was used to detect differential gene expression after compound treatment (Appendix A) [28], and the twenty most significantly enriched pathways were identified by KEGG enrichment analysis [29]. The MAPK signaling pathway was the most significantly enriched pathway in both **1**-treated cells and **1a**-treated cells vs. controls cells (Figure 4A). The MAPK family is a group of serine/threonine protein kinases that mediate intracellular signaling and are important transmitters of signals from the cell surface to the interior of the nucleus. The MAPK signaling pathway consists of three main MAPK subfamilies, including MAPK14, the c-Jun N-terminal kinase or stress-activated protein kinase (JNK or SAPK), and the extracellular signal-regulated kinases (ERK MAPK, Ras/Raf1/MEK/ERK). There is increasing evidence that activation of the ERK MAPK pathway is involved in the pathogenesis and progression of human cancers and is one of the most important pathways for cell proliferation [30].

To further confirm the role of the MAPK signaling pathway in the mechanisms of action of **1** and **1a** on cells, we examined the protein expression levels of MEK and ERK and their phosphorylation levels of key factors in the MAPK signaling pathway by Western blotting. Western blotting results showed that treatment with **1** and **1a** reduced p-MEK and p-ERK expression in a concentration-dependent manner (Figure 4B,C). These results revealed an antagonistic relationship between **1**/**1a** and MAPK signaling in HEL cells. In addition, we found that both **1** and **1a** significantly decreased the protein levels of c-MET, an upstream target protein of the MAPK signaling pathway [31].

### 2.4. Compounds **1** and **1a** Antagonize the MAPK Signaling Pathway to Induce Apoptosis by Inhibiting c-MET Protein Expression

c-MET is an upstream protein of the MAPK signaling pathway, and it has been reported in the literature that c-MET activates Ras through the GRB2-SOS complex [32], which in turn activates RAF and activates the downstream MAPK pathway. In addition, the binding of the receptor c-MET to hepatocyte growth factor (HGF) activates several signaling pathways, such as the PI3K/AKT, NF-κB and STAT3/5 pathways [15]. Thus, the overexpression of HGF and the overactivation of c-MET play key roles in tumorigenesis [33]. Another study showed that there is high expression of c-MET and its sole ligand HGF in HEL cells [34], and in view of this, it is reasonable to suspect that in HEL cells, **1** and **1a** induce apoptosis by mediating c-MET activation of the MAPK signaling pathway. Therefore, we analyzed the protein expression of c-MET after compound treatment of HEL cells. The experiments showed that both **1** and **1a** reduced the expression levels of c-MET protein in a time-dependent and dose-dependent manner (Figure 5A,B). This suggests that the expression of c-MET was suppressed after treatment with the compounds, and the RNA-seq analysis also showed that the PI3K/AKT/mTOR signaling pathway, a downstream signaling pathway of c-MET, was one of the most significantly changed signaling pathways in cells treated with **1** or **1a** vs. control cells (Figure 4A) [35]. We speculate that the alteration of the MAPK signaling pathway is likely to be mediated by c-MET. To test this hypothesis, we further validated the protein expression levels of the c-MET downstream pathway components PI3K/AKT/mTOR and JAK2/STAT3. As expected, the Western blotting results showed that both **1** and **1a** reduced the protein expression levels of p-PI3K, p-mTOR, and p-STAT3 in a dose-dependent manner (Figure 5C,D). This result further confirms that the molecular mechanism by which **1** and **1a** act on HEL cells may be mediated through c-MET.

### 2.5. Compounds **1** and **1a** Suppress the MAPK, PI3K/AKT, and STAT3 Signaling Pathways by Decreasing the mRNA and Protein Levels of c-MET

To confirm the mechanism by which the downstream signaling pathways MAPK, PI3K/AKT, and STAT3 are mediated by **1** and **1a** via c-MET, overexpression of c-MET was performed in HEL cells (Appendix A, Appendix A) [36]. According to Western blotting validation experiments, the decrease in the protein levels of c-MET caused by treatment with **1** and **1a** could be attenuated by MET transient transfection, indicating the essential role of MET in the antiproliferative activity of these compounds. This result was verified by assessment of the protein expression levels of p-MEK, p-ERK, p-mTOR, and p-STAT3, which are key targets in the downstream signaling pathway (Figure 6A,B). To further explore how c-MET protein expression levels were affected by **1** and **1a**, we examined the mRNA expression levels of c- MET by RT q-PCR after compound treatment [37]. Intriguingly, c-MET mRNA levels in HEL cells were reduced in a dose-dependent manner with compound treatment, indicating that c-MET expression is subject to pretranscriptional regulation (Figure 6C); this indicates that c-MET and its downstream pathways are repressed, at least in part, through c-MET pretranscriptional regulation after treatment with **1** and **1a**.

These data indicated that in HEL cells, **1** and **1a** exert their cytostatic effects at least in part by decreasing the mRNA and protein levels of c-MET and subsequently inhibiting the MAPK, PI3K/AKT, and STAT3 signaling pathways as shown in Figure 7.

In this study, we found that both **1** and **1a** effectively decreased the viability of HEL in leukemic cells in a dose-dependent manner, and **1** and **1a** mainly decreased c-MET mRNA and protein levels in a time-dependent and dose-dependent manner, thereby inhibiting the phosphorylation of its downstream signaling components MAPK, PI3K/mTOR, and STAT3 and subsequently inducing apoptosis. Our study clarified that the molecular mechanisms of compound **1a** were the same as those of lead compound **1** and that c-MET was involved in the regulatory effects of **1** and **1a** on signaling pathways, including the MAPK, PI3K/mTOR, and STAT3 pathways, which is a new antitumor mechanism of **1**. This study provides a solid theoretical basis for the development and utilization of **1** in tumor therapy and provides more drug candidates for the treatment of leukemia and a theoretical basis for applying such treatments. Certainly, there are some limitations to this research. For example, the mechanism by which **1** and **1a** decrease the mRNA level of c-MET needs to be further explored; this will be addressed in future studies.

## 3. Materials and Methods

### 3.1. Cell Lines and Culture

For this study, the HEL cell line was obtained from the American Type Culture Collection (ATCC, Manassas, VA, USA), and the cells were cultured in 1640 medium (Gibco, Grand Island, NYC, New York, NY, USA) supplemented with 100 U/mL penicillin, 0.1 mg/mL streptomycin, 0.25 µg/mL amphotericin B, and 10% heat-inactivated fetal bovine serum, at 37 °C in a humidified atmosphere with 5% CO_2_.

### 3.2. Cell Viability Assays

A total of 5 × 10^3^ cells were added to each well of the 96-well plate and treated with 0.1625, 0.3125, 0.625, 1.25, 2.5, and 5 μM test compounds or DMSO (Beijing Solarbio Science & Technology Co., Ltd. Beijing, China) for 48 h. Then, 10 μL of MTT (5 mg/mL, in PBS) (Beijing Solarbio Science & Technology Co., Ltd. Beijing, China) or CCK-8 (Beijing Solarbio Science & Technology Co., Ltd. Beijing, China) was added into each well and further incubated for 4 h. For MTT, the samples were centrifuged at 2000 r/min for 15 min, the supernatant was removed, and 10 μL of DMSO was added to each well. Then, the cells were incubated for 10 min, and the absorbance value was measured at 490 nm using a microplate reader (Bio-Tek, Winooski, VT, USA). For CCK8, after incubation, the absorbance of the plates was measured directly at 450 nm.

### 3.3. Apoptosis Analysis

HEL cells were incubated in 6-well plates at 4 × 10^5^ per well and after 24 h different concentrations of compounds were added, and the incubation was continued for 6 h. Then, the cells were collected and washed twice with PBS. The cells were stained with reagents from an Annexin V-FITC/PI apoptosis kit (Shanghai Yishan Biotech Co., Ltd. Shanghai, China, AP001). Apoptosis was determined by flow cytometry (ACEA Biosciences, San Diego, CA, USA).

### 3.4. Hoechst 33258 Staining

HEL cells were incubated in 6-well plates at 4 × 10^5^ cells/well and cultured for 24 h. Then, the cells were incubated in the presence of different concentrations of compounds for 6 h. According to the Hoechst 33258 kit (Beijing Solarbio Science & Technology Co., Ltd., Beijing, China, C0020) instructions, Hoechst 33258 staining solution was added directly to the sample; the sample was incubated for 25 min and then gently washed twice with PBS. Finally, cell morphology was assessed under a fluorescence microscope (Leica, Wetzlar, Hessian, Germany).

### 3.5. TUNEL Staining

After fixation of the cells with 4% paraformaldehyde, the TUNEL staining procedure was performed according to the instructions of YF^®^ 488 TUNEL Kit (US Everbright^®^ Inc. Suzhou, China, T6013S), the DAPI dye solution used was purchased from Beyotime Institute of Biotechnology (Beyotime Biotech Inc. Shanghai, China, P0131), and the cells were then observed under a fluorescence microscope (Leica, Wetzlar, Hessian, Germany). The number of TUNEL-positive cells and total cells in each field was determined. The apoptosis rate was shown as the average percentage of TUNEL-positive cells among the total cells in the three fields selected for each section.

### 3.6. Western Blotting

HEL cells were treated with different concentrations of **1**, **1a,** or 0.1% DMSO for 6 h and collected. After washing twice with PBS, cells were treated with a RIPA lysis buffer (Beyotime Biotech Inc. Shanghai, China) solution containing protease inhibitors to extract the total proteins and then the protein concentration was determined by BCA protein assay (Solarbio Science Technology, Beijing, China, PC0020). The proteins (40–60 μg per lane) were separated using 10% SDS-PAGE (Beyotime Biotech Inc. Shanghai, China), wet-transferred to PVDF membranes, and blotted with primary antibodies specific for GAPDH (Proteintech, Chicago, Illinois, USA, #21800-1-AP), c-MET (Proteintech, Chicago, Illinois, USA, 25869-1-AP), MEK (Proteintech, Chicago, Illinois, USA, 11049-1-AP), p-MEK (Cell Signaling Technology, Inc, Boston, MA, USA, #914), ERK1/2 (Proteintech, 16443-1-AP), p-ERK (Cell Signaling Technology, Inc, Boston, MA, USA, #4370S), STAT3 (Proteintech, Chicago, IL, USA, 10253-2-AP), p-STAT3 (Cell Signaling Technology, Inc, Boston, Massachusetts, USA, #9145), p-PI3K (Affinity, AF3242), PI3K (Affinity, Melbourne, Victoria, Australia, AF5112), mTOR (Proteintech, Chicago, IL, USA, 2065AP), p-mTOR (Cell Signaling Technology, Inc, Boston, Massachusetts, USA, #5536), PARP (Cell Signaling Technology, Inc, Boston, MA, USA, #9532), and cleaved PARP (Cell Signaling Technology, Inc, Boston, MA, USA, #94885) at 4 °C. After 12 h, the membranes were washed three times with TBST, and then HRP-conjugated secondary antibodies were applied for 2 h. The previous cleaning procedure was repeated. Later, color development was performed with Smart-ECL Enhanced reagent (Changzhou Smart-Lifesciences Biotechnology Co., Ltd., Changzhou, China), and band intensities were quantified using Image Lab 5.1 (Bio-Rad Laboratories Inc, Hercules, CA, USA) and normalized to those of GAPDH.

### 3.7. RNA-Sequencing (RNA-Seq) Experiment and Analysis

After treatments, RNA samples were extracted and sequenced by Metware Biotechnology Co., Ltd. (Wuhan, China), and the cDNA libraries were sequenced on the Illumina sequencing platform by Metware Biotechnology Co., Ltd. (Wuhan, China). Bioinformatics analysis was performed on the Metware Cloud platform (Metware Biotechnology Co., Ltd., Wuhan, China).

### 3.8. Overexpression of c-MET

The MET (NM_000245) plasmid and its lentivirus were generated by GeneChem (Shanghai GeneChem CO., Ltd., Shanghai, China). The GV513 vector was used to construct a full-length c-MET plasmid and to complete the packaging of the lentivirus with Helper 1.0 and Helper 2.0 (Shanghai GeneChem CO., Ltd., Shanghai, China). Then, transfection of the lentivirus was conducted using HitransG A (Shanghai Genechem CO., Ltd., Shanghai, China), and the negative control virus CON335 (Ubi-MCS-CBh-gcGFP-IRES-puromycin, provided by Shanghai Genechem CO., Ltd., Shanghai, China) was used as a negative control virus. Cells were collected for further experiments 24 h after transfection, following the manufacturer’s recommendations. The sequences are listed in the additional Appendix A.

### 3.9. Real-Time q-PCR

After treatments, total RNA was extracted from cells using the RNA isolation reagent TRIzol (Beijing Solarbio Science & Technology Co., Ltd., Beijing, China) and transcribed into cDNA using Hifair^®^ III 1st Strand cDNA Synthesis SuperMix for q-PCR (gDNA digester plus, Yeasen Biotechnology (Shanghai) Co., Ltd., Shanghai, China). Then, cDNA was used for qRT-PCR analysis with Hieff UNICON^®^ Universal Blue q-PCR SYBR Green Master Mix (Yeasen Biotechnology (Shanghai) Co., Ltd., Shanghai, China), and the reaction mixtures were incubated for 2 min at 95 °C followed by 40 cycles of 10 s at 95 °C and 30 s at 60 °C. The 2^−ΔΔ*C*t^ method was used to calculate relative gene expression, and the mRNA level of the gene of interest was normalized to that of GAPDH. The primers used for the q-PCR were as follows:

c-MET, forward 5′-AGCAATGGGGAGTGTAAAGAGG-3′,

reverse 5′-CCCAGTCTTGTACTCAGCAAC-3′;

GAPDH, forward 5′-CATCAAGAAGGTGGTGAAGCA-3′;

reverse 5′-TCAAAGGTGGAGGAGTGGGT-3′.

### 3.10. Statistical Analysis

All data were obtained from three independent experiments. Statistical analysis was performed using GraphPad Prism 8 (GraphPad Software, La Jolla, CA, USA). Student’s *t* test was applied to assess the significance of differences between two groups. The data are displayed as the mean ± SD, and the threshold for significance was set to *p* < 0.05.

## Figures and Tables

**Figure 1 ijms-24-08113-f001:**
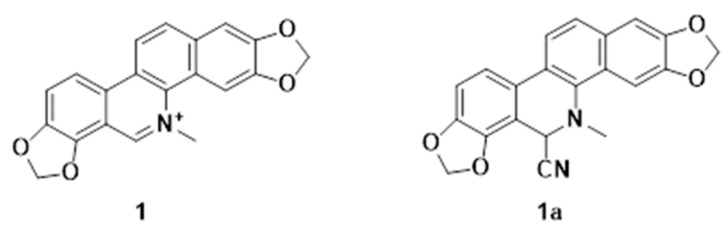
Structures of Sanguinarine and its cyanide derivative.

**Figure 2 ijms-24-08113-f002:**
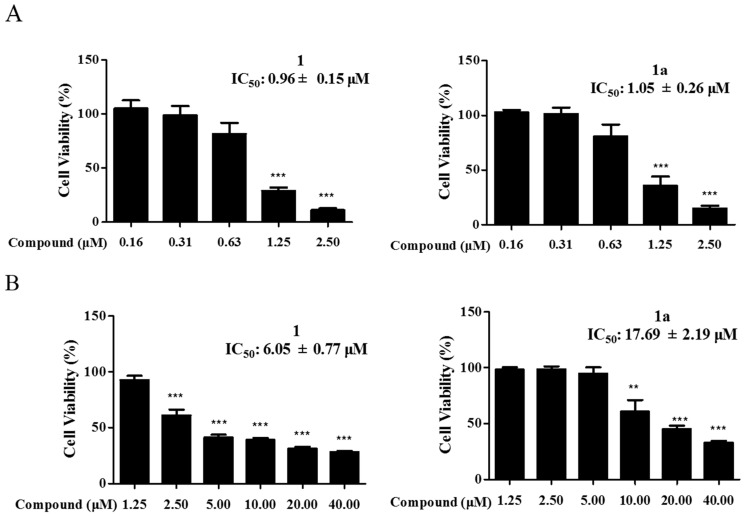
Compounds **1** and **1a** inhibit the proliferation of HEL cells and LX-2 cells. (**A**) HEL cells were incubated with **1** and **1a**. (**B**) LX-2 cells were incubated with **1** and **1a**. Cell activity was assessed using the CCK-8 kit. Data were collected from three independent experiments. Error bars represent standard deviation. Two-tailed *p* values were derived from unpaired Student’s *t* test. “**”, *p* < 0.01; “***”, *p* < 0.001.

**Figure 3 ijms-24-08113-f003:**
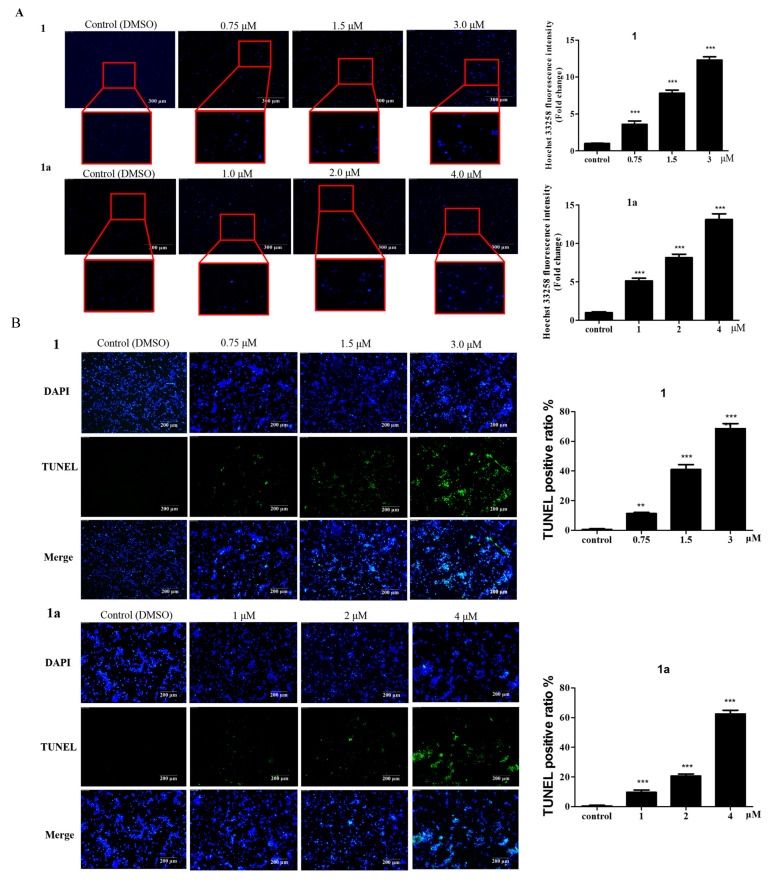
The effect of **1** and **1a** on cell apoptosis. (**A**) Hoechst 33258 staining of HEL cells. Bright blue fluorescence increases in a dose-dependent manner Scale bar: 300 μm (n = 3). (**B**) TUNEL staining and the average data of HEL cells. Scale bar: 200 μm (n = 3). (**C**) Different concentrations of **1** or **1a** were used to treat HEL cells. The rate of apoptosis was assessed with an Annexin V-FITC/PI kit. (**D**) The expression levels of apoptosis-related proteins from HEL cells treated with **1** and **1a** were analyzed by Western blotting. “*”, *p* < 0.05; “**”, *p* < 0.01; “***”, *p* < 0.001.

**Figure 4 ijms-24-08113-f004:**
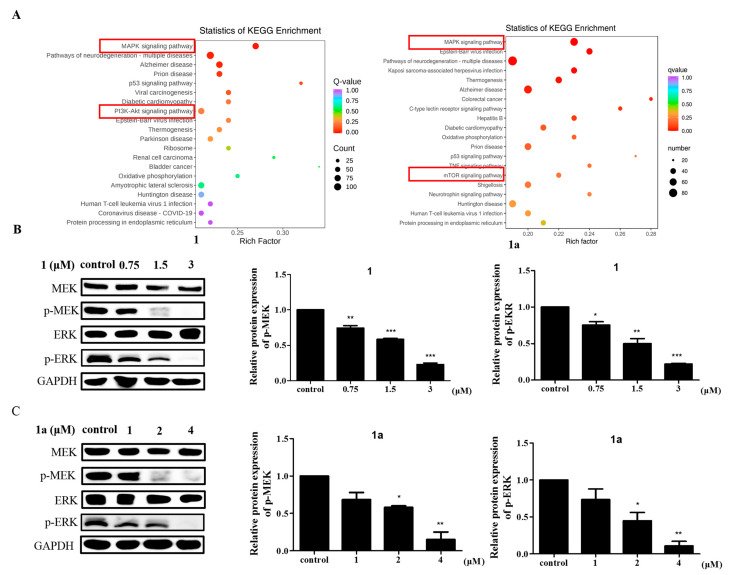
Compounds **1** and **1a** inhibit HEL cell activity by blocking MAPK signaling pathways. (**A**) RNA-seq was performed in HEL cells pretreated with DMSO, **1** (3 μM), or **1a** (4 μM), and KEGG enrichment analysis was used to analyze the most significantly enriched signaling pathways. The vertical coordinate represents the KEGG pathway, and the horizontal coordinate represents the rich factor. The larger the rich factor is, the greater the enrichment level. The larger the dot is, the greater the number of differential genes enriched in the pathway. The redder the color of the dot is, the more significant the enrichment. (**B**,**C**) The expression of intrinsic MAPK-signaling-pathway-related proteins in HEL cells was measured by Western blotting. “*”, *p* < 0.05; “**”, *p* < 0.01; “***”, *p* < 0.001.

**Figure 5 ijms-24-08113-f005:**
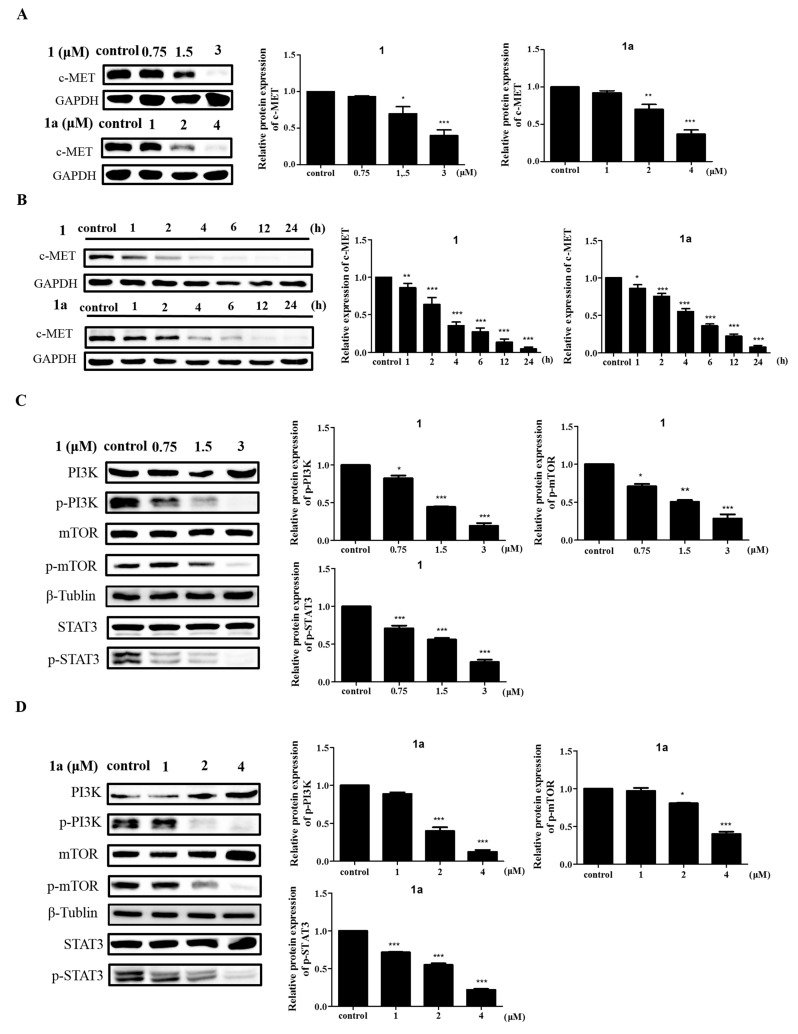
Compounds **1** and **1a** induce HEL cell apoptosis via the c-MET signaling pathway. (**A**) Both **1** and **1a** decrease c-MET protein expression in a dose-dependent manner. (**B**) Both **1** and **1a** decrease c- MET protein expression in a time-dependent manner. (**C**,**D**) Both **1** and **1a** decrease the phosphorylation levels of downstream proteins in the c-MET signaling pathway in a dose-dependent manner. “*”, *p* < 0.05; “**”, *p* < 0.01; “***”, *p* < 0.001.

**Figure 6 ijms-24-08113-f006:**
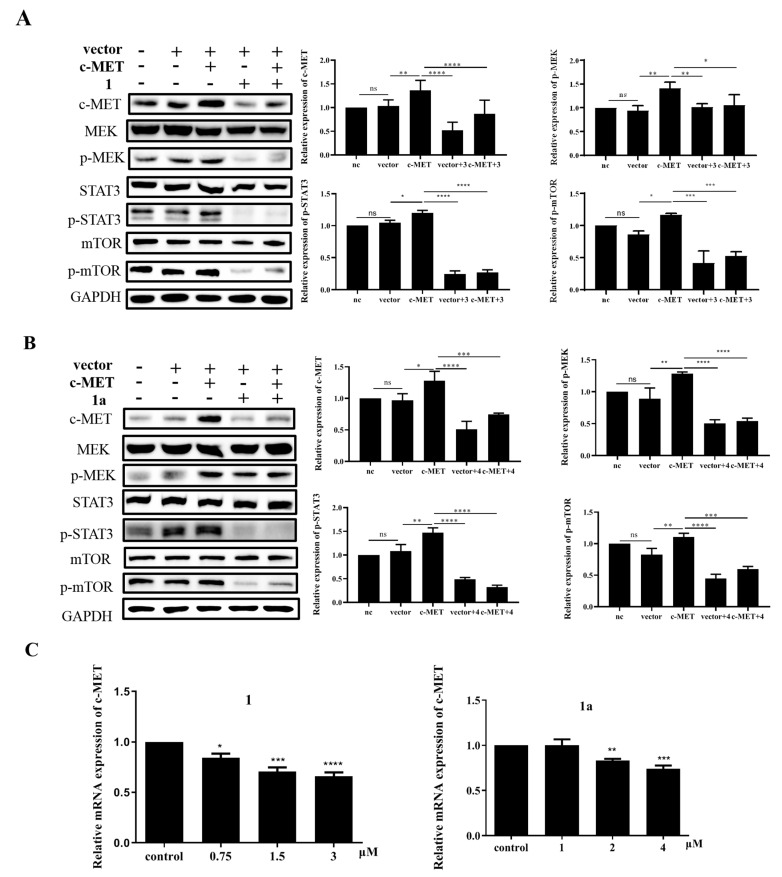
Compounds **1** and **1a** decrease the mRNA and protein levels of c-MET. (**A**, **B**) Overexpression of c-MET by lentiviral infection of HEL cells and detection of c-MET-signaling-pathway-related protein expression by Western blotting. (**C**) After treatment with **1** and **1a**, the mRNA levels of c-MET were detected by RT q-PCR. “ns”, no significance; “*”, *p* < 0.05; “**”, *p* < 0.01; “***” *p* < 0.001, “****”, *p* < 0.0001.

**Figure 7 ijms-24-08113-f007:**
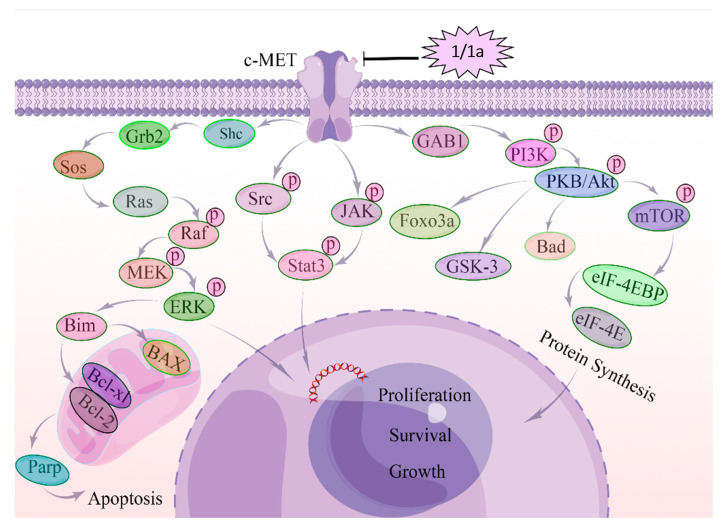
Molecular mechanism of c-MET regulating HEL cells after the treatment with compounds **1** and **1a**. c-MET is overactivated in human erythroid leukemia cell cells (HEL). After the treatment with **1** and **1a**, the mRNA expression level and protein expression level of c-MET were suppressed, and subsequently, the c-MET homodimerization and autophosphorylation were affected. Then, GAB-1, Shc, Grb-2, and STAT3 were suppressed. Grb2 inhibits SOS, SOS suppresses RAS, and then RAS inhibits RAF, MEK, and ERK/MAPK. Suppressed ERK/MAPK can modulate transcription factors to regulate cell behaviors. Suppressed GAB-1 inhibits AKT, PKB, and mTOR to regulate transcription factors.

## Data Availability

The data of RNA-seq from this study were submitted to the NCBI database under SRA, ID: SUB12932460.

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
