# Peer review of "Cytostatic Activity of Sanguinarine and a Cyanide Derivative in Human Erythroleukemia Cells Is Mediated by Suppression of c-MET/MAPK Signaling"

_ijms, 2023, doi:10.3390/ijms24098113_

Round 1

Reviewer 1 Report

The research article titled" Cytostatic activity of sanguinarine and a cyanide derivative in 1 human erythroleukemia cells is mediated by suppression of c- 2 MET/MAPK signaling" is a good study. Some of my major concerns are

1. There are two bands in ERK 1/2. The authors have shown only 1 band for ERK and p-ERK in Fig 4B,C.

2. Although it is not necessary, but TUNEL assay provides excellent images for cell undergoing cell death.

3. Line number 151 Veractivation, Does the authors mean overactivation.

4.  The authors need to revised the manuscript once again for some small grammatical mistakes. 

5. This is a good study and well performed experiments supported by the results. 

2. 

Author Response

Response to Reviewer 1’s Comments

Dear Reviewer 1,

Thank you very much for your time involved in reviewing our manuscript and we hope that the explanation has fully addressed all of your concerns. According to your suggestion, our manuscript has been revised. The modification traces were highlighted with the yellow marker in our revised manuscript, and response to Reviewer 1 is as the following:

Point 1: There are two bands in ERK 1/2. The authors have shown only 1 band for ERK and p-ERK in Fig 4B,C.

Response 1: (1) According to the ERK1/2 (Proteintech, 16443-1-AP) antibody manual, it shows that the product can be observed as one band (as shown in the figure below), and we considered that the design of this manufacturer's product is as one band, so it was also one band in our experimental results.

(2) In the raw data of our experiment(as shown in the figure below), we can observe two bands. Because of the difference in the expression of the two bands in HEL cells resulted in a more pronounced expression of the band with molecular weight of 42 KD compared to the band with 44 KD at the same exposure level, and the small spacing between the two bands due to the concentration of the separation gel during the experiment, both of which resulted in the presentation of results that were not suitable to be observed. We hope that the presentation of the original bands will clear up your doubts.

44 KD

42 KD

48 KD

35 KD

63 KD

Point 2: Although it is not necessary, but TUNEL assay provides excellent images for cell undergoing cell death.

Response 2: In our revised manuscript, the results of TUNEL assay of 1 and 1a was shown in Fig3B.

Point 3: Line number 151 Veractivation, Does the authors mean overactivation.

Response 3: We would like to thank you for carefully reading our manuscript. In our revised manuscript, the word has been modified. The change have been marked with the highlighted words.

Point 4: The authors need to revised the manuscript once again for some small grammatical mistakes

Response 4: We apologize for the language problems in the manuscrirt. We have carefully checked and improved the English writing in our revised manuscript. Meanwhile, we have invited American Journal Experts to make linguistic and grammatical corrections to the manuscript (ID: V5Q7MCNV). All changes have been marked with the highlighted words.

Point 5: This is a good study and well performed experiments supported by the results.

Response 5: We would like to take this opportunity to express our sincere gratitude for your time and efforts in reviewing my article. Your positive feedback and encouragement have been instrumental in enhancing the quality of the final manuscript, and we are truly honored to have had the benefit of your expertise and insights.

Thank you very much for your suggestions again and we hope that this revision will be recognized by you.

Reviewer 2 Report

Summary

Xu et al., have investigated the anti-leukemic activity of sanguinarine (1) and structurally modified sanguinarine (1a) in the manuscript entitled, “Cytostatic activity of sanguinarine and a cyanide derivative in human erythroleukemia cells is mediated by suppression of c- MET/MAPK signaling”. Authors claim that the structurally modified derivative of sanguinarine has lower cytotoxicity compared to the sanguinarine and is evident from the results in Fig 2. The mechanism of action of compound 1 and 1a via MAPK pathway has been elucidated in this study. c-MET receptor, a target for tumor treatment acts upstream to MAPK and activates the signaling by phosphorylating downstream effectors. Both compounds 1 and 1a show reduced the protein expression levels of p-PI3K, p-mTOR and p-STAT3 in a dose-dependent manner in HEL cells. Authors further show decreased c-MET mRNA and protein levels suppresses the MAPK, PI3K/AKT, and STAT3 signaling pathways. Although the data in the manuscript is relevant to tumor therapy, the exact molecular mechanism by which these two compounds decrease the c-MET mRNA and protein expression is unknown. Authors could further elucidate the molecular mechanism of compounds 1 and 1a to strengthen the current version of manuscript. It would be great if the authors provide information on wound healing and cell migration assays.

 Conceptual comments

The manuscript is clear, comprehensive and of relevance to the cancer field. The flow of the manuscript is smooth. Important articles in the recent past have been included and justified in the manuscript. There are no excessive self-citations. In order to use the compound in cancer treatment it is important to reduce the cell toxicity. Authors have done admiring task of structural optimization of sanguinarine to reduce its toxicity without changing its cytostatic activity. The data provided in the current manuscript is of great importance, relevance and a valuable addition to the existing knowledge of the scientific community.

Specific comments

What was the specific purpose of using LX2 cell line?

Please provide the scale bar in Fig3A. Were the images quantified? Please provide quantification data. The resolution of Fig 3A and 3B is poor. The x and y axis labels cannot be read well. Please include images with better resolution.

RNA seq data analysis information in supplementary is Table S2. In the 2.3 section main text it is mistakenly labelled as Fig S2. Please rectify.

Was the data in FigS2 showing c-MET overexpression of c-MET in HEL cells quantified? There is no visual change in the expression of c-MET is observed. Please provided quantification data.

I could not find FigS3 in the supplementary data file but line number 179 in the main text discusses about FigS3.

Line number 151- what is "veractivation" of c-MET? I think it is over activation c-MET? If yes, please correct the typo.

Providing a brief legend for Figure 7 would be useful.

Author Response

Response to Reviewer 2’s Comments

Dear Reviewer 2,

We feel great thanks for your professional review work on our manuscript. As you are concerned, our manuscript need to be improved. According to your suggestions, our manuscript has been revised. The modification traces were highlighted with the yellow marker in our revised manuscript, and response to Reviewer 2 is as the following:

Point 1: Summary:

Xu et al., have investigated the anti-leukemic activity of sanguinarine (1) and structurally modified sanguinarine (1a) in the manuscript entitled, “Cytostatic activity of sanguinarine and a cyanide derivative in human erythroleukemia cells is mediated by suppression of c- MET/MAPK signaling”. Authors claim that the structurally modified derivative of sanguinarine has lower cytotoxicity compared to the sanguinarine and is evident from the results in Fig 2. The mechanism of action of compound 1 and 1a via MAPK pathway has been elucidated in this study. c-MET receptor, a target for tumor treatment acts upstream to MAPK and activates the signaling by phosphorylating downstream effectors. Both compounds 1 and 1a show reduced the protein expression levels of p-PI3K, p-mTOR and p-STAT3 in a dose-dependent manner in HEL cells. Authors further show decreased c-MET mRNA and protein levels suppresses the MAPK, PI3K/AKT, and STAT3 signaling pathways. Although the data in the manuscript is relevant to tumor therapy, the exact molecular mechanism by which these two compounds decrease the c-MET mRNA and protein expression is unknown. Authors could further elucidate the molecular mechanism of compounds 1 and 1a to strengthen the current version of manuscript. It would be great if the authors provide information on wound healing and cell migration assays.

Response 1: (1) Your comments are pertinent and important, but due to current time constraints and experimental conditions, the follow-up study could not be continued. We will definitely study this opinion in more depth when we have the opportunity in the future. (2) Thanks for your suggestion. The wound healing and cell migration assays are indeed important in assessing the ability of cells to migrate. However, unfortunately, HEL cells are semi-suspended and semi-adherent cells, a property that prevents the cells from fully adhering to the wall in the wound healing assay and cell migration assay and thus does not yield the desired experimental results, and we are sorry that we cannot provide the relevant experimental results.

Point 2: Conceptual comments:

The manuscript is clear, comprehensive and of relevance to the cancer field. The flow of the manuscript is smooth. Important articles in the recent past have been included and justified in the manuscript. There are no excessive self-citations. In order to use the compound in cancer treatment it is important to reduce the cell toxicity. Authors have done admiring task of structural optimization of sanguinarine to reduce its toxicity without changing its cytostatic activity. The data provided in the current manuscript is of great importance, relevance and a valuable addition to the existing knowledge of the scientific community.

Response 2: Thank you very much for your affirmation and approval to our manuscript.

Specific comments

Point 3: What was the specific purpose of using LX2 cell line?

Response 3: Thank you for your question. Since sanguinarine (1) is a compound with hepatotoxicity [1], it is essential to examine the toxicity of compounds to normal hepatocytes, and LX-2 cells have been reported to be valuable new tools in the study of liver disease [2]. Therefore, the LX-2 cell line was selected as the cell line for 1 and 1a hepatotoxicity evaluation in our manuscript.

[1] Kosina P, Walterová D, Ulrichová J, Lichnovský V, Stiborová M, Rýdlová H, Vicar J, Krecman V, Brabec MJ, Simánek V. Sanguinarine and chelerythrine: assessment of safety on pigs in ninety days feeding experiment. Food Chem Toxicol. 2004. 42 (1): 85-91. doi: 10.1016/j.fct.2003.08.007.

[2] Xu L, Hui AY, Albanis E, Arthur MJ, O'Byrne SM, Blaner WS, Mukherjee P, Friedman SL, Eng FJ. Human hepatic stellate cell lines, LX-1 and LX-2: new tools for analysis of hepatic fibrosis. Gut. 2005, 54 (1): 142-51. doi: 10.1136/gut.2004.042127.

Point 4: Please provide the scale bar in Fig3A. Were the images quantified? Please provide quantification data. The resolution of Fig3A and 3B is poor. The x and y axis labels cannot be read well. Please include images with better resolution.

Response 4: (1) According to your comment, we have provided quantification data and the scale bar in the revised manuscript and Fig3A. (2) We are very sorry for the poor resolution of the previously provided images, and in our revised manuscript, we have provided a better resolution for Fig3.

Point 5: RNA seq data analysis information in supplementary is Table S2. In the 2.3 section main text it is mistakenly labelled as Fig S2. Please rectify.

Response 5: We were really sorry for our careless mistakes. In our revised manuscript, the wrong mark has been modified and marked in red and highlighted.

Point 6: Was the data in FigS2 showing c-MET overexpression of c-MET in HEL cells quantified? There is no visual change in the expression of c-MET is observed. Please provided quantification data.

Response 6: According to your comment, we have quantified the fluorescence expression of HEL cells after our overexpression assay (as shown in the blow).

negative control group (NC): Normal HEL cells, c-MET overexpression group: Lentivirus (packaging plasmid vector and c-MET gene sequence), vector group: Lentivirus (packaging plasmid vector), the vector group is to exclude the influence of vector. And we provide the cellular status of different groups as well as c-MET protein expression levels after lentiviral overexpression (as shown in the blow). Corresponding modifications can also be found in the supplementary material and Figure S2.

Point 7: I could not find FigS3 in the supplementary data file but line number 179 in the main text discusses about FigS3.

Response 7: We are very sorry for our careless mistake. Due to our mistake, we wrote FigS2 as FigS3, and in our revised manuscript we have corrected FigS3 to FigS2.

Point 8: Line number 151- what is "veractivation" of c-MET? I think it is over activation c-MET? If yes, please correct the typo.

Response 8: Yes, we would like to thank you for carefully reading our manuscript. In our revised manuscript, the word has been modified and the change have been marked with highlighted words.

Point 9: Providing a brief legend for Figure 7 would be useful.

Response 9: We sincerely appreciate the valuable comments. We have provided a brief legend for Figure 7, you will see our changes in our revised manuscript and Figure 7, which has been alternatively marked with the highlighted.
